# A Method to Track Moving Targets Using a Doppler Radar Based on Converted State Kalman Filtering

**Xian Zhao, Xuanzhi Zhao \*, Zengli Liu and Wen Zhang**

Faculty of Information Engineering and Automation, Kunming University of Science and Technology, Kunming 650500, China; 20212104011@stu.kust.edu.cn (X.Z.); lzl@kust.edu.cn (Z.L.); 20222204054@stu.kust.edu.cn (W.Z.)
\* Correspondence: zhaoxuanzhi@kust.edu.cn

**Abstract:** Strong nonlinearity between Doppler measurement and target motion in Doppler radar target tracking leads to the inadequate utilization of measurement information and limited tracking accuracy. We solved this problem by combining converted state Kalman filtering and the Interacting Multiple Model. This maneuvering target tracking method is suitable for Doppler measurement. First, we converted the target motion in the Cartesian coordinate to the polar coordinate. Then, we expanded the measurement equation to include Doppler measurement, making target motion linearly related to the Doppler radar observation vectors and allowing efficient utilization of measurement information. Next, we used unscented transformation to calculate the statistical characteristics of the process noise in the polar coordinate. This process helps to reduce the noise error caused by the coordinate system transformation in the original converted state Kalman filter. Finally, the system effectively tracks targets that may perform maneuvers with unknown motion during actual tracking. Using the converted state Kalman filter with Doppler measurement as a sub-filter, an Interacting Multiple Model tracking method can be constructed to adjust the model probabilities without going through nonlinear transformation. Simulation results show that the technique can achieve effective target tracking in Doppler measurement application scenarios and has higher tracking accuracy in non-maneuvering and maneuvering scenarios.

**Keywords:** target tracking; converted state; unscented transformation; Doppler measurement; interacting multiple model





## 1. Introduction

The Doppler radar is known for its strong anti-interference and clutter suppression capabilities due to its ability to remove noise in the frequency domain [1]. Owing to its significant benefits, the Doppler radar is widely used in areas like target tracking [2], battlefield surveillance, and traffic control. When tracking a target, a Doppler radar can obtain the range and bearing of the target, the target's Doppler measurement, or the target's radial velocity. As a result, Doppler measurement is the only measurement value that contains target speed information among all Doppler radar measurements. Studies have shown that Doppler measurement can effectively improve target tracking accuracy [3]. However, physical quantities expressed in the Cartesian coordinate are usually used to describe the target motion when tracking moving targets using radar measurements. The range, bearing, and Doppler measurement provided by Doppler radar are all polar coordinate systems, making it difficult to eliminate the nonlinearity between the target motion and the radar measurement. As a result, using Doppler radar measurements can be challenging due to the nonlinearity introduced by the coordinate system transformation between the polar and Cartesian systems. Since the Doppler measurement is a composite function of multiple Cartesian state variables, it has strong nonlinearity, making it challenging to utilize the Doppler measurement [4] efficiently. Therefore, the central research theme of Doppler radar

target tracking is to seek reasonable methods to resolve the nonlinearity between target motion and measurement.

One study [5] proposes the Sequential Extended Kalman Filter (SEKF) for the target tracking problem using Doppler measurement, where the measurement conversion Kalman filter was first used to filter the position measurement linearly. Then, the Extended Kalman Filter (EKF) [6–8] was used to process the Doppler measurement. However, discarding high-order terms above second order during the EKF linearization process can lead to more significant errors when dealing with strong nonlinearity. Some studies, such as [9,10], extend the Debiased Converted Measurement (DCM) [11–13] and the Unbiased Converted Measurement (UCM) methods [14–16]. These models are only considered for position measurement to solve the nonlinear target tracking problem using Doppler measurement. These authors have deduced a nonlinear relationship between Doppler measurements and target states by constructing Doppler pseudo-measurements from range and Doppler measurements. Another study [17] proposed the Sequential Unscented Kalman Filter (SUKF), which uses UKF [18–21] first to perform decorrelation processing on the range and Doppler measurements with measurement error correlation and process the position measurement and pseudo-Doppler measurement sequentially. Another study [22] also proposed the Statically Fused Converted Measurement Kalman Filters. In this model, the DCM Kalman filter is first used to process the position measurement and estimate the target position state. Then, the standard Kalman filter is used to calculate the pseudo-Doppler state. Finally, the minimum variance estimates the static fusion position and pseudo-Doppler state. Another researcher [23] improved the model presented by [22], using UCM and DCM to process position measurement and Doppler pseudo-measurement, respectively. Their model aims to make the estimation results free of bias. One researcher [24] also proposed the Converted Measurement Kalman Filtering algorithm with Range Rate (CMKFRR) using this method to convert range, bearing, and Doppler measurement to position and velocity in the Cartesian coordinate system. While this conversion is unbiased and consistent, the measurement conversion requires prior knowledge of the distribution of lateral velocities. Another study [25] proposed a Decorrelated Unbiased Converted Measurement Kalman Filter with Range rate (DUCMKF-R). This method aims to produce unbiased and consistent filtering results by calculating the covariance of the converted measurement error based on the predicted value. However, the above methods still require pseudo-measurements to reduce the nonlinearity or decorrelation of measurement errors. These methods show that unbiased filtering cannot use accurate Doppler measurements for filtering updates. Although these methods have improved the target tracking performance of the Doppler radar, improving tracking accuracy is still necessary. One study [26] proposed the Converted State Kalman Filter (CSKF) algorithm to address the nonlinear problems in the motion and measurement equations in target tracking. This algorithm converts the equations of motion to the polar coordinate using the Cartesian coordinate, making the state and observation linearly related. As a result, nonlinear filtering can be transformed into a standard problem that can be processed using linear Kalman filtering. However, challenges remain, such as finding an effective method suitable for Doppler measurement and dealing with complex maneuvering scenarios with unknown target motion.

This study proposes a novel target-tracking algorithm called IMM-CSKF-D. This algorithm combines the Converted State Kalman Filter with Doppler measurement (CSKF-D) and the Interacting Multiple Model (IMM) [27–29] to address some of the existing problems in the target tracking algorithm. The proposed algorithm utilizes Doppler measurement and derives the measurement equation. Then, the unscented transform (UT) is derived when the process noise is converted from the Cartesian coordinate to the polar coordinate to calculate the converted noise's statistical characteristics. The process also ensures that the calculation results are highly accurate and that the CSKF-D algorithm is obtained. Then, we combined the CSKF-D and IMM to address the situation where the target has maneuvering motion during the actual tracking process. Afterward, we used the CSKF-D as a sub-filter of the IMM to propose the IMM-CSKF-D target tracking algorithm. Simulation results show

that the proposed algorithm outperforms other tracking algorithms that utilize Doppler measurement based on its higher estimation accuracy.

## 2. Description of Problem

### 2.1. Decomposition of Motion

When fusing multi-source information, different fusion spaces lead to different fusion performances. In the traditional CSKF algorithm, the state equation in Cartesian coordinates is converted to polar coordinates to ensure consistency with the measurement filtering. As a result, the state equation is derived from polar coordinates. Then, the transformation of process noise is analyzed.

In establishing the target motion equation in the polar coordinate, the motion equation in the Cartesian coordinate is converted to polar coordinates. This mechanism is achieved by orthogonally decomposing the Cartesian coordinate system velocity $V$ into the radial velocity $\dot{r}$ and the tangential velocity $v_\theta$. This process is depicted in Figure 1.

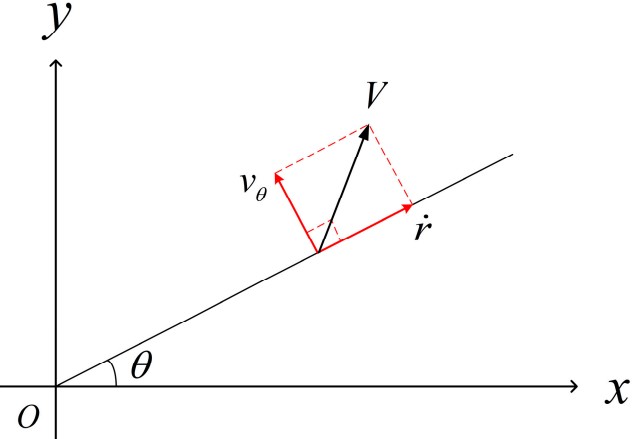

**Figure 1.** Decomposition of a motion state.

### 2.2. Target Motion Equations in the Polar Coordinate System

2.2.1. CV Motion

If a target moves in a straight line at a constant speed, its motion process is described as constant velocity (CV) motion. In the polar coordinate, the state equation of CV motion is as follows:

$$X_{cv}(k+1) = \boldsymbol{\Phi}_{cv}(k)X_{cv}(k) + \boldsymbol{\Gamma}_{cv}(k)W_{cv}(k) \tag{1}$$

where $X_{cv}(k) = [\theta(k) \quad \dot{\theta}(k) \quad r(k) \quad \dot{r}(k)]^T$ represents the target state at time $k$ during CV motion; $\theta(k)$ and $r(k)$ are the bearing and the range, respectively; $\dot{\theta}(k)$ and $\dot{r}(k)$ are the angular velocity and radial velocity, respectively; and $W_{cv}(k)$ is process noise.

$$\boldsymbol{\Phi}_{cv}(k) = \begin{bmatrix} 1 & T & 0 & 0 \\ 0 & 1 - T\frac{\dot{r}(k)}{r(k)} & 0 & 0 \\ 0 & 0 & 0 & T \\ 0 & 0 & 0 & 1 \end{bmatrix} \tag{2}$$

where $\boldsymbol{\Phi}_{cv}(k)$ is the state transition matrix and $T$ is the sampling period.

$$\boldsymbol{\Gamma}_{cv}(k) = \begin{bmatrix} 0 & -\frac{T}{r(k)} & 0 & 0 \\ 0 & 0 & \frac{1}{2}T^2 & T \end{bmatrix}^T \tag{3}$$

where $\boldsymbol{\Gamma}_{cv}(k)$ is the process noise-driven matrix.

2.2.2. CA Motion

If the target moves in a straight line with constant acceleration, its motion is described as constant acceleration (CA). In the polar coordinate, the state equation of CA motion is as follows:

$$\boldsymbol{X}_{ca}(k+1) = \boldsymbol{\Phi}_{ca}(k)\boldsymbol{X}_{ca}(k) + \boldsymbol{\Gamma}_{ca}(k)\boldsymbol{W}_{ca}(k) \tag{4}$$

where $\boldsymbol{X}_{ca}(k) = [\theta(k) \quad \dot{\theta}(k) \quad \ddot{\theta}(k) \quad r(k) \quad \dot{r}(k) \quad \ddot{r}(k)]^T$ represents the target state at time $k$ during CA motion; $\theta(k)$ and $r(k)$ are the bearing and the range, respectively; $\dot{\theta}(k)$ and $\dot{r}(k)$ are the angular velocity and radial velocity, respectively; $\ddot{\theta}(k)$ and $\ddot{r}(k)$ are the angular acceleration and radial acceleration, respectively; and $\boldsymbol{W}_{ca}(k)$ is process noise.

$$\boldsymbol{\Phi}_{ca}(k) = \begin{bmatrix} 1 & T & 0 & 0 & 0 & 0 \\ 0 & 1 & T & 0 & 0 & 0 \\ 0 & -\frac{1}{T}\frac{\ddot{r}(k)}{r(k)} & 1 - 2T\frac{\dot{r}(k)}{r(k)} & 0 & 0 & 0 \\ 0 & 0 & 0 & 1 & T & \frac{1}{2}T^2 \\ 0 & 0 & 0 & 0 & 1 & T \\ 0 & 0 & 0 & 0 & 0 & 1 \end{bmatrix} \tag{5}$$

where $\boldsymbol{\Phi}_{ca}(k)$ is the state transition matrix.

$$\boldsymbol{\Gamma}_{ca}(k) = \begin{bmatrix} 0 & \frac{T}{r(k)} & 0 & 0 & 0 & 0 \\ 0 & 0 & 0 & \frac{1}{6}T^3 & \frac{1}{2}T^2 & T \end{bmatrix}^T \tag{6}$$

where $\boldsymbol{\Gamma}_{ca}(k)$ is the process noise-driven matrix.

## 3. Converted State Kalman Filtering with Doppler Measurement

*3.1. Measurement Equations with Doppler Measurement*

We used a model developed in one study [26] to introduce Doppler measurement and expand the observation equation. Assuming that the radar position is at the coordinate origin in the polar coordinate, the radar measurement equation with Doppler measurement is expressed as follows:

$$\begin{aligned} \boldsymbol{Z}(k) &= [\ \theta_m(k) \quad r_m(k) \quad \dot{r}_m(k)]^T \\ &\begin{cases} \theta_m(k) = \arctan\frac{y(k)}{x(k)} + \widetilde{\theta}(k) \\ r_m(k) = \sqrt{x^2(k) + y^2(k)} + \widetilde{r}(k) \\ \dot{r}_m(k) = \frac{x(k)\dot{x}(k) + y(k)\dot{y}(k)}{\sqrt{x^2(k) + y^2(k)}} + \widetilde{\dot{r}}(k) \end{cases} \end{aligned} \tag{7}$$

where $x(k)$, $y(k)$, $\dot{x}(k)$, and $\dot{y}(k)$ are the accurate positions and velocities of the target in the $X$ and $Y$ directions in the Cartesian coordinate; $\theta_m(k)$, $r_m(k)$ and $\dot{r}_m(k)$ are the bearing measurement, range measurement, and Doppler measurement, respectively. Moreover, $\widetilde{\theta}(k)$, $\widetilde{r}(k)$, and $\widetilde{\dot{r}}(k)$ are the corresponding measurement noises. Assuming that these noises are zero-mean Gaussian white noise with variances of $\sigma_\theta^2$, $\sigma_r^2$, and $\sigma_{\dot{r}}^2$, respectively, $\sigma_r^2$ and $\sigma_\theta^2$ are uncorrelated; $\sigma_\theta^2$ and $\sigma_{\dot{r}}^2$ are uncorrelated; and $\sigma_\theta^2$ and $\sigma_{\dot{r}}^2$ are correlated with the correlation coefficient $\rho$, developed using the following equation:

$$\mathrm{cov}[\widetilde{r}(k), \widetilde{\dot{r}}(k)] = \rho\sigma_r\sigma_{\dot{r}}. \tag{8}$$

Then, the measurement noise covariance at time $k$ is expressed as follows:

$$\boldsymbol{R}(k) = \begin{bmatrix} \sigma_\theta^2 & 0 & 0 \\ 0 & \sigma_r^2 & \rho\sigma_r\sigma_{\dot{r}} \\ 0 & \rho\sigma_r\sigma_{\dot{r}} & \sigma_{\dot{r}}^2 \end{bmatrix}. \tag{9}$$

According to the state Equations (1) and (4) and the radar measurement Equation (7), the converted target state has a linear relationship with the measurement vector. The measurement equations of CV motion and CA motion are as follows:

$$
\begin{aligned}
\mathbf{Z}(k) &= \mathbf{H}_{cv}\mathbf{X}_{cv}(k) + \mathbf{V}(k) \\
\mathbf{Z}(k) &= \mathbf{H}_{ca}\mathbf{X}_{ca}(k) + \mathbf{V}(k)
\end{aligned}
\tag{10}
$$

where $\mathbf{H}_{cv} = \begin{bmatrix} 1 & 0 & 0 & 0 \\ 0 & 0 & 1 & 0 \\ 0 & 0 & 0 & 1 \end{bmatrix}$, $\mathbf{H}_{ca} = \begin{bmatrix} 1 & 0 & 0 & 0 & 0 & 0 \\ 0 & 0 & 0 & 1 & 0 & 0 \\ 0 & 0 & 0 & 0 & 1 & 0 \end{bmatrix}$, and $\mathbf{V}(k)$ is the measurement error.

### 3.2. Analysis of Process Noise

Process noise is crucial in introducing randomness to the target's motion and is a vital component of the state equation. Therefore, we must use a process noise transformation method to transform process noise from Cartesian to polar coordinate systems. The UT is quite beneficial in accurately estimating nonlinear systems without the need for Jacobian derivation. It can also help transform a series of sample points to approximate the posterior probability density of the state. As a result, this study uses UT transformation to calculate the mean and covariance of process noise in the polar coordinate.

In the Cartesian coordinate, the process noise is modeled using zero-mean Gaussian white noise, transformed into the radial and tangential directions in the polar coordinate system using a rotation matrix. The process noise conversion equations of CV motion and CA motion are as follows:

$$
\begin{aligned}
\mathbf{W}_{cv}(k) &= \begin{bmatrix} \dot{v}_\theta(k) \\ \ddot{r}(k) \end{bmatrix} = \begin{bmatrix} -\sin\theta(k) & \cos\theta(k) \\ \cos\theta(k) & \sin\theta(k) \end{bmatrix} \begin{bmatrix} \dot{v}_x(k) \\ \dot{v}_y(k) \end{bmatrix} \\
\mathbf{W}_{ca}(k) &= \begin{bmatrix} \ddot{v}_\theta(k) \\ \dddot{r}(k) \end{bmatrix} = \begin{bmatrix} -\sin\theta(k) & \cos\theta(k) \\ \cos\theta(k) & \sin\theta(k) \end{bmatrix} \begin{bmatrix} \ddot{v}_x(k) \\ \ddot{v}_y(k) \end{bmatrix}
\end{aligned}
\tag{11}
$$

where $\dot{v}_\theta(k), \ddot{v}_\theta(k), \ddot{r}(k)$, and $\dddot{r}(k)$ are the tangential process noise and radial process noise of CV motion and CA motion, respectively. Additionally, $\dot{v}_x(k), \dot{v}_y(k), \ddot{v}_x(k)$, and $\ddot{v}_y(k)$ are the process noise of CV motion and CA motion in the *X* direction and *Y* direction of the Cartesian coordinate, respectively. However, these noises are not correlated.

We calculate the statistical characteristics of the process noise converted to the polar coordinate using the UT transformation as follows:

(1) The $2n + 1$ sigma sample points (*n* is the state dimension) are generated based on the mean and variance of the three-dimensional random vector $\begin{bmatrix} \theta(k) & \dot{v}_x(k) & \dot{v}_y(k) \end{bmatrix}^T$ (CV motion) or $\begin{bmatrix} \theta(k) & \ddot{v}_x(k) & \ddot{v}_y(k) \end{bmatrix}^T$ (CA motion).

(2) The sigma sample points are then substituted into Equation (13) to calculate the sample points generated via nonlinear mapping.

$$
\begin{aligned}
f(\begin{bmatrix} \theta(k) & \dot{v}_x(k) & \dot{v}_y(k) \end{bmatrix}^T) &= \begin{bmatrix} -\sin\theta(k) & \cos\theta(k) \\ \cos\theta(k) & \sin\theta(k) \end{bmatrix} \begin{bmatrix} \dot{v}_x(k) \\ \dot{v}_y(k) \end{bmatrix} \\
f(\begin{bmatrix} \theta(k) & \ddot{v}_x(k) & \ddot{v}_y(k) \end{bmatrix}^T) &= \begin{bmatrix} -\sin\theta(k) & \cos\theta(k) \\ \cos\theta(k) & \sin\theta(k) \end{bmatrix} \begin{bmatrix} \ddot{v}_x(k) \\ \ddot{v}_y(k) \end{bmatrix}
\end{aligned}
\tag{12}
$$

(3) Through the weighted sum, we obtained the mean and variance of the process noise $\begin{bmatrix} \dot{v}_\theta(k) & \ddot{r}(k) \end{bmatrix}^T$ (CV motion) or $\begin{bmatrix} \ddot{v}_\theta(k) & \dddot{r}(k) \end{bmatrix}^T$ (CA motion) in the polar coordinate system.

### 3.3. CSKF Algorithm with Doppler Measurement

We used the above process to construct the time-varying target state equation containing Doppler measurements in the polar coordinate, deriving the statistical characteristics of the related process noise. As a result, the proposed CSKF-D algorithm can achieve real-time

fusion using a specific time-varying state equation each time. The specific steps of the algorithm are as follows:

Filter input: $X(k,k)$, $P(k,k)$, $Z(k+1)$
Filter output: $X(k+1,k+1)$, $P(k+1,k+1)$

(1)   We predict the system state and covariance.

$$X(k+1,k) = \boldsymbol{\Phi}(k)X(k,k)$$
$$P(k+1,k) = \boldsymbol{\Phi}(k)P(k,k)\boldsymbol{\Phi}(k)^T + \boldsymbol{\Gamma}(k)Q(k)\boldsymbol{\Gamma}(k)^T \tag{13}$$

(2)   Then, Kalman gain is calculated.

$$K(k+1) = P(k+1,k)H^T[R(k) + HP(k+1,k)H^T]^{-1} \tag{14}$$

(3)   Finally, we update the system state and covariance.

$$X(k+1,k+1) = X(k+1,k) + K(k+1)[Z(k+1) - HX(k+1,k)]$$
$$P(k+1,k+1) = P(k+1,k) - K(k+1)[R(k) + HP(k+1,k)H^T]K(k+1)^T \tag{15}$$

where $Q(k)$ is the process noise covariance matrix.

## 4. Target Tracking with Combined CSKF-D and IMM Method

In target tracking, the target movement is complex, changeable, and unknown when the target makes maneuvering motions. As a single-model filter, the Kalman filter often faces challenges in achieving optimal tracking results when the target makes maneuvering motions. However, due to its multi-model nature, IMM can overcome the problem of significant estimation error in single-model filtering. Typically, IMM uses two or more models to describe the possible motion states of the target during the tracking process and fuses the filtering results under different motion models through probability weighting to obtain a more accurate target motion estimate.

As a result, we propose the IMM-CSKF-D algorithm, which combines CSKF-D and IMM, using CSKF-D as a sub-filter of IMM. The IMM-CSKF-D algorithm is conducted recursively, and each recursion step includes four steps: input interaction, state filtering, model probability update, and state fusion output. Figure 2 depicts the IMM-CSKF-D algorithm flow.

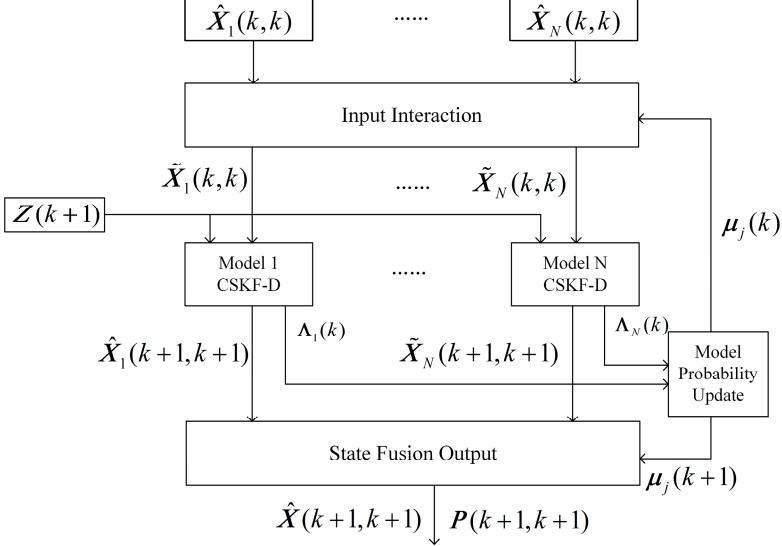

**Figure 2.** IMM-CSKF-D algorithm flow chart.

The recursion steps of the IMM-CSKF-D algorithm containing $N$ models from time $k$ to time $k+1$ are presented as follows:

(1)   Input interaction:

$$\widetilde{X}_j(k,k) = \sum_{i=1}^{N} \hat{X}_i(k,k)\mu_{ij}(k+1,k)$$
$$\widetilde{P}_j(k,k) = \sum_{i=1}^{N} \mu_{ij}(k+1,k)\Big\{P_i(k,k) + [\hat{X}_i(k,k) - \widetilde{X}_j(k,k)][\hat{X}_i(k,k) + \widetilde{X}_j(k,k)]\Big\} \tag{16}$$

where $\mu_{ij}(k+1,k) = \frac{p_{ij}\mu_i(k)}{\overline{c}_j}$ is the mixed transition probability; $\overline{c}_j = \sum_{i=1}^{N} p_{ij}\mu_i(k)$, where $\mu_i(k)$ represents the model probability of model $i$ of the target at time $k$; and $p_{ij}$ represents the transition probability from model $i$ to model $j$. $\hat{X}_i(k,k)$ and $P_i(k,k)$, respectively, represent the state estimate of the target model $i$ at time $k$ and its covariance matrix. $\widetilde{X}_j(k,k)$ and $\widetilde{P}_j(k,k)$, respectively, represent the state interaction value of the target model $j$ at time $k$ and its covariance matrix.

(2)   State filtering: $\widetilde{X}_j(k,k)$ and $\widetilde{P}_j(k,k)$ are used as filter inputs to obtain the state estimate $\hat{X}_j(k+1,k+1)$ and covariance matrix $P_j(k+1,k+1)$ at the next moment. Section 3.3 of this manuscript describes the single model filtering algorithm process.

(3)   Model probability update:

$$\mu_j(k+1) = \frac{\Lambda_j(k+1)\overline{c}_j}{c} \tag{17}$$

where $c = \sum_{i=1}^{N} \Lambda_j(k+1)\overline{c}_j$ is the normalization constant.

$$\Lambda_j(k+1) = \frac{1}{\sqrt{|2\pi S_j(k+1)|}} \exp[-\tfrac{1}{2}v_j^T(k+1)S_j^{-1}(k+1)v_j(k+1)]$$
$$v_j(k+1) = Z(k+1) - H(k)\hat{X}_i(k+1,k)$$
$$S_j(k+1) = H(k)P_i(k+1,k)H(k)^T + R(k) \tag{18}$$

where $\hat{X}_i(k+1,k)$ and $P_i(k+1,k)$ are the predicated state and covariance of the target at time $k+1$; $v_j(k+1)$ and $S_j(k+1)$ are the measurement residuals and their covariances. Therefore, our model is directly derived from the difference between observations and linear predictions in the residual calculation. As a result, our model is free of nonlinear approximation errors, which makes it capable of yielding more accurate model probabilities.

(4)   State fusion output: Based on the posterior probability of each model, a probability-weighted summation of the state estimates of each filter obtains the final estimated state and covariance estimate.

$$\hat{X}(k+1,k+1) = \sum_{j=1}^{N} \hat{X}_j(k+1,k+1)\mu_j(k+1)$$
$$P(k+1,k+1) = \sum_{j=1}^{N} \mu_j(k+1)\Big\{P_j(k+1,k+1) + [\hat{X}_j(k+1,k+1)$$
$$-\hat{X}(k+1,k+1)][\hat{X}_j(k+1,k+1) + \hat{X}(k+1,k+1)]\Big\} \tag{19}$$

## 5. Simulation Results and Analysis

We simulated the proposed method using MATLAB to verify its performance. We compared the CSKF-D algorithm proposed with SEKF [5], SUKF [17], CMKFRR [24], and DUCMKF-R [25]. All algorithms included Doppler measurements, and 300 Monte Carlo simulations were conducted under the same conditions. The performance evaluation index uses the target position's root mean square errors (Position RMSE) and velocity's root mean square errors (Velocity RMSE). The results showed that the CSKF-D algorithm performed excellently when the target has multiple motion states.

### 5.1. CV Model

We analyzed the target performing CV motion in the two-dimensional space. The target's initial position was (10 km, 10 km), and the initial velocity was (8 m/s, 10 m/s). The Doppler radar located at the origin of the coordinates provides the range, bearing, and Doppler measurements of the target at a sampling period of 1 s. The standard deviations

of the measurement noise are $\sigma_r$, $\sigma_\theta$, and $\sigma_{\dot{r}}$, respectively. The correlation coefficient was $\rho = 0.5$. We also set up two simulation scenarios with different noise variances to analyze the tracking performance of the CSKF-D algorithm at different measurement noises. Table 1 presents the parameters, and Figures 3 and 4 depict the corresponding simulation results.

**Table 1.** Measurement noise parameters in two scenarios.

| Scenario | $\sigma_r$ (m) | $\sigma_\theta$ (deg) | $\sigma_{\dot{r}}$ (m/s) |
|----------|----------------|------------------------|---------------------------|
| 1 | 50 | 0.5 | 0.05 |
| 2 | 100 | 1 | 0.1 |

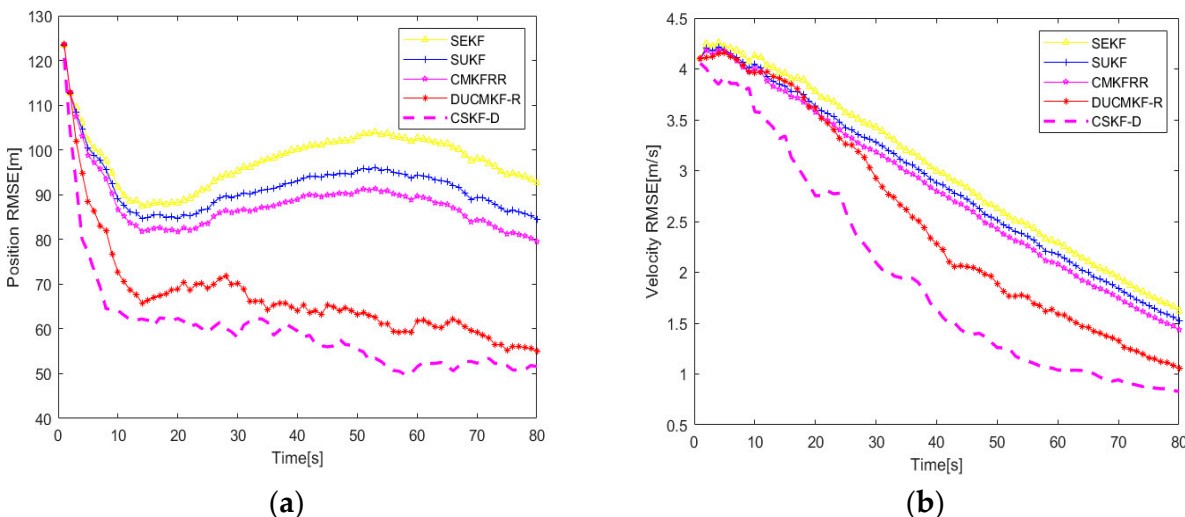

**Figure 3.** RMSEs in CV motion (scenario 1). (**a**) RMSE of position. (**b**) RMSE of velocity.

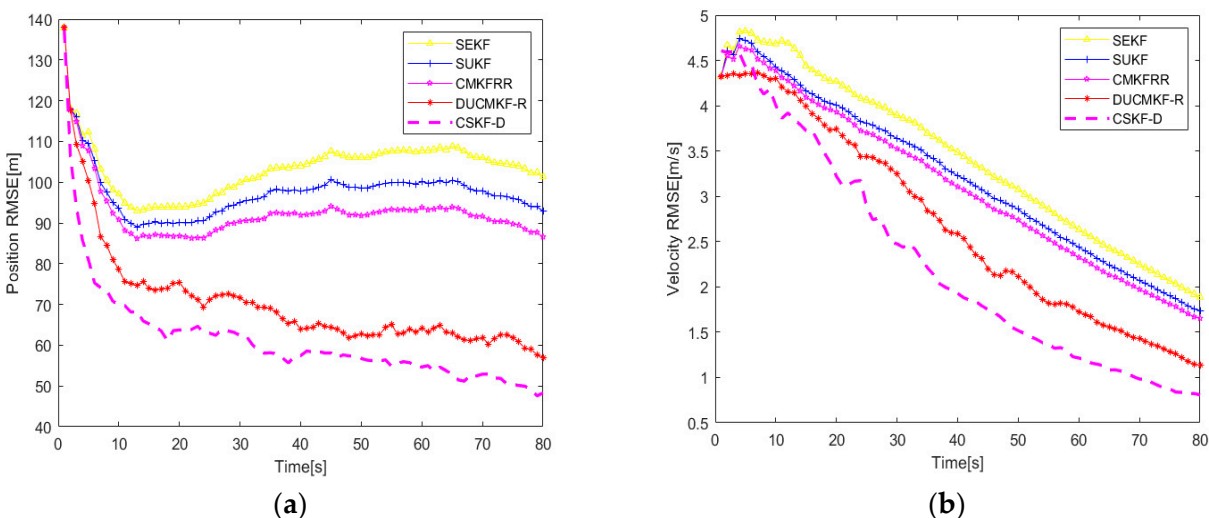

**Figure 4.** RMSEs in CV motion (scenario 2). (**a**) RMSE of position. (**b**) RMSE of velocity.

Figures 3 and 4 show these algorithms' estimation errors of CV motion under different measurement noises. Table 2 shows the mean values of Position RMSE and Velocity RMSE for all methods. From the results, our proposed CSKF-D algorithm outperformed other algorithms due to its higher estimation accuracy and faster convergence rate. The Position RMSEs for the SEKF, SUKF, and CMKFRR algorithms rise after declining initially, while the CSKF-D algorithm and DUCMKF-R algorithm only rise slightly and continue declining. As for the Velocity RMSE, the CSKF-D algorithm only maintains a small gap with the other algorithms in the initial stage and quickly widens the gap, showing excellent performance.

**Table 2.** Performance comparison in CV scenarios.

| Measurement Noise Parameters | Method | RMSE of Position (m) | RMSE of Velocity (m/s) |
|---|---|---|---|
| $\sigma_r = 50$ m $\sigma_\theta = 0.5$ deg $\sigma_{\dot{r}} = 0.05$ m/s | SEKF | 99.14 | 2.87 |
| | SUKF | 93.96 | 2.69 |
| | CMKFRR | 89.35 | 2.53 |
| | DUCMKF-R | 72.74 | 2.37 |
| | CSKF-D | 61.86 | 1.83 |
| $\sigma_r = 100$ m $\sigma_\theta = 1$ deg $\sigma_{\dot{r}} = 0.1$ m/s | SEKF | 111.24 | 3.44 |
| | SUKF | 102.96 | 3.26 |
| | CMKFRR | 97.62 | 3.17 |
| | DUCMKF-R | 75.54 | 2.97 |
| | CSKF-D | 62.33 | 2.34 |

The SEKF algorithm can produce more significant errors when dealing with solid nonlinearities like those found in Doppler measurement because of its limitation in discarding high-order terms above second order during the linearization process. As a result, SUKF uses a series of deterministic samples to approximate the posterior probability density of the state, reaching at least the second-order approximation, resulting in more accurate filtering than SEKF. The proposed CSKF-D algorithm can process state and measurement vectors using a linear Kalman filter, ensuring dynamic estimation convergence. When conducting fuse in polar coordinates, this method has a smaller variance than Cartesian coordinates, making the CSKF-D algorithm better in estimation accuracy and greater in convergence rate than CMKFRR and its improved algorithm DUCMKF-R.

### 5.2. CA Model

In the CA case, the target's initial position was (10 km, 10 km), the initial velocity was (8 m/s, 10 m/s), and the initial acceleration was (2 m/s², 2 m/s²). The other parameters are the same as those in Section 5.1. Figures 5 and 6 show the corresponding simulation results.

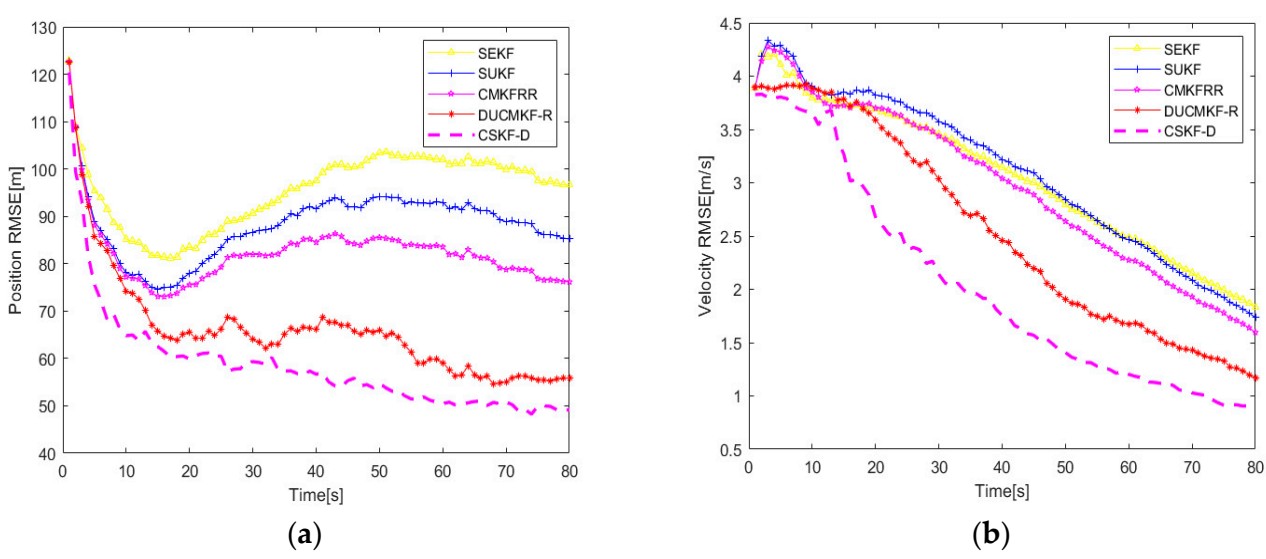

(**a**)  (**b**)

**Figure 5.** RMSEs in CA motion (scenario 1). (**a**) RMSE of position. (**b**) RMSE of velocity.

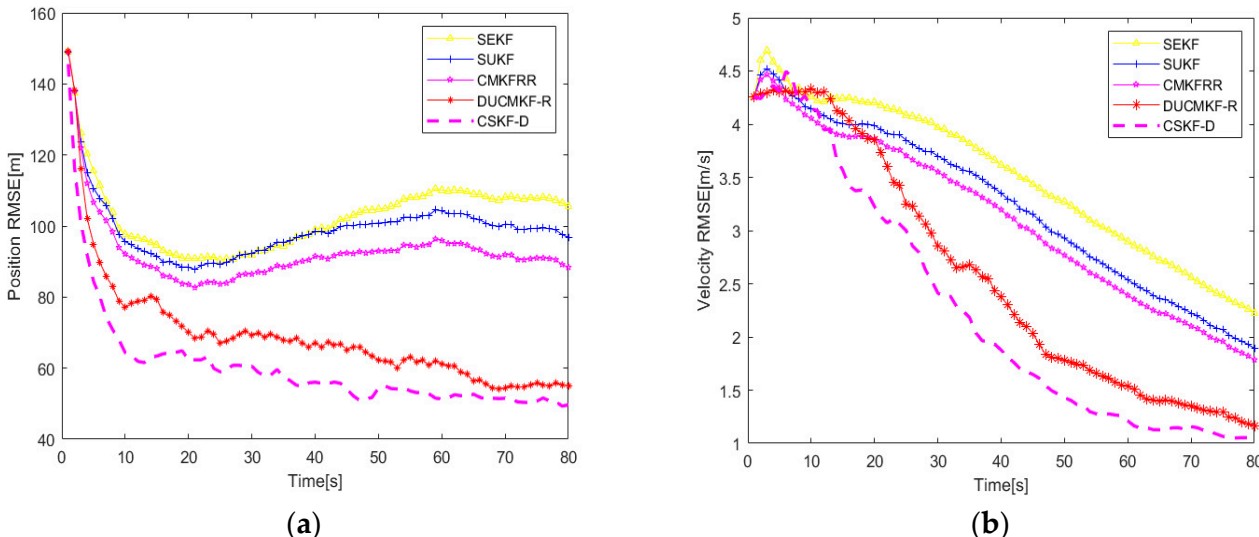

**Figure 6.** RMSEs in CA motion (scenario 2). (**a**) RMSE of position. (**b**) RMSE of velocity.

Figures 5 and 6 demonstrate the estimation errors of CA motion based on various algorithms at different measurement noises. Table 3 shows the mean values of Position RMSE and Velocity RMSE for all methods. The CSKF-D algorithm outperforms the other filtering algorithms regarding estimation accuracy and rate of convergence in the CA model. The success of the CSKF-D algorithm could be attributed to using a more appropriate coordinate system to construct the state, which helps avoid severe non-Gaussian distortion in information fusion.

**Table 3.** Performance comparison in CA scenarios.

| Measurement Noise Parameters | Method | RMSE of Position (m) | RMSE of Velocity (m/s) |
|---|---|---|---|
| $\sigma_r = 50$ m $\sigma_\theta = 0.5$ deg $\sigma_{\dot{r}} = 0.05$ m/s | SEKF | 100.54 | 2.94 |
| | SUKF | 92.41 | 2.73 |
| | CMKFRR | 89.23 | 2.61 |
| | DUCMKF-R | 71.85 | 2.42 |
| | CSKF-D | 61.17 | 1.97 |
| $\sigma_r = 100$ m $\sigma_\theta = 1$ deg $\sigma_{\dot{r}} = 0.1$ m/s | SEKF | 112.53 | 3.50 |
| | SUKF | 103.69 | 3.24 |
| | CMKFRR | 98.50 | 3.21 |
| | DUCMKF-R | 79.54 | 2.89 |
| | CSKF-D | 62.87 | 2.57 |

We conducted simulations to compare the conversion of nonlinear coordinates between Cartesian coordinate and polar coordinate systems. The goal was to elucidate the reason for the performance advantage of the proposed method. We consider the CV motion as an example: a four-dimensional vector in Cartesian coordinates and a three-dimensional vector in polar coordinates. The conversion equations are as follows:

(1)   From polar coordinates to Cartesian coordinates, we deduce the following:

$$\boldsymbol{\Psi} = \begin{bmatrix} x & y & \dot{x} & \dot{y} \end{bmatrix}^T = \begin{bmatrix} r\cos\theta & r\sin\theta & \dot{r}\cos\theta - v_\theta\sin\theta & \dot{r}\sin\theta + v_\theta\cos\theta \end{bmatrix}^T + w_c. \quad (20)$$

(2)   The following equation is deduced from Cartesian coordinates to polar coordinates:

$$\boldsymbol{\Phi} = \begin{bmatrix} r & \theta & \dot{r} \end{bmatrix}^T = \begin{bmatrix} \sqrt{x^2 + y^2} & \arctan\left(\frac{y}{x}\right) & \frac{x\dot{x} + y\dot{y}}{\sqrt{x^2 + y^2}} \end{bmatrix}^T + w_p \quad (21)$$

where $x$, $y$, $\dot{x}$, and $\dot{y}$ are the valid positions and velocities, respectively, in the Cartesian coordinate system. Moreover, $r$, $\dot{r}$, $\theta$, and $v_\theta$ are the true range, radial velocity, bearing, and tangential velocity, respectively, in the polar coordinate system. $w_c$ and $w_p$ are the zero-mean Gaussian noises with covariances $Q_c$ and $Q_p$, respectively.

Then, two-dimensional vectors are selected from the two coordinate systems for conversion and for comparing the simulation results. A low-dimensional vector is used to map a high-dimensional vector. Table 4 shows the selection of scenarios. Figure 7 depicts the simulation results.

**Table 4.** Parameters of three scenarios.

|  | Scenario 1 | Scenario 2 | Scenario 3 |
|---|:---:|:---:|:---:|
| Cartesian coordinates | $x, \dot{x}$ | $x, y$ | $y, \dot{y}$ |
| Polar coordinates | $r, \dot{r}$ | $r, \theta$ | $\theta, \dot{r}$ |

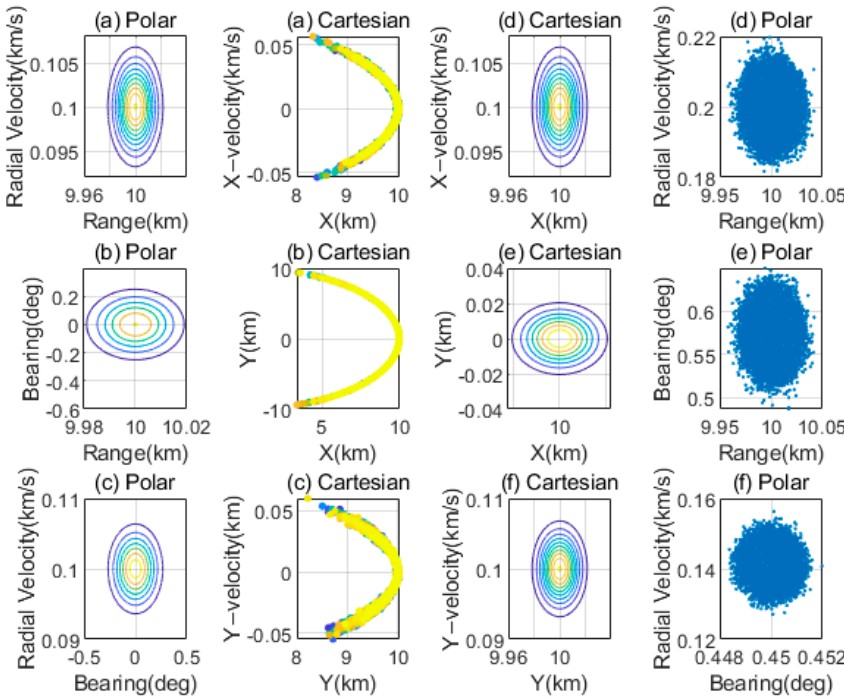

**Figure 7.** Distribution before and after two conversions.

As shown in Figure 7, (a), (b), and (c) convert the Gaussian distribution in the Cartesian coordinates of three scenarios to polar coordinates. After conversion, (a), (b), and (c) showed significant bending distortion, while (d), (e), and (f) showed reverse conversion. Despite conversion, they exhibit Gaussian distribution, indicating that the transformed distribution is likely. We found that using the stronger Gaussian likelihood resulted in a smaller information loss caused when using fusion based on this approximation. The results also showed that fusion in polar coordinates causes smaller errors. Therefore, we proposed a method that can operate solely in polar coordinates, which helps avoid the fundamental shortcomings of measurement conversion methods.

### 5.3. IMM Model

The CSKF-D algorithm is suitable for tracking targets in scenarios involving no maneuvering motion. However, maneuvering scenarios with unknown motion patterns are more common in real-world target tracking. Combining the CSKF-D algorithm with IMM to address this challenge can result in more accurate model probability estimation. Additionally, the CSKF-D algorithm used in the sub-filter offers higher estimation accuracy, allowing

for high-precision maneuvering target tracking. We considered a specific scenario where the target was initially located at (10 km, 10 km), travelled at an initial velocity of (8 m/s, 10 m/s), and had an initial acceleration of (1 m/s², 1 m/s²). The target's range, bearing, and Doppler measurements were measured at a sampling period of 1 s using a Doppler radar located at the coordinate origin. The standard deviations of the measurement noises for these parameters are $\sigma_r$, $\sigma_\theta$, and $\sigma_{\dot{r}}$, respectively. In addition, the correlation coefficient was $\rho = 0.5$. We simulated two measurement noises with different statistical characteristics. Table 5 shows the measurement noise variances.

**Table 5.** Measurement noise parameters in two scenarios.

| Scenario | $\sigma_r$ (**m**) | $\sigma_\theta$ (**deg**) | $\sigma_{\dot{r}}$ (**m/s**) |
|:---:|:---:|:---:|:---:|
| 1 | 60 | 0.6 | 0.06 |
| 2 | 120 | 1.2 | 0.12 |

We compared our proposed IMM-CSKF-D algorithm with the IMM-SEKF, IMM-SUKF, IMM-CMKFRR, and IMM-DUCMKF-R algorithms. The simulation results are shown in Figures 8 and 9.

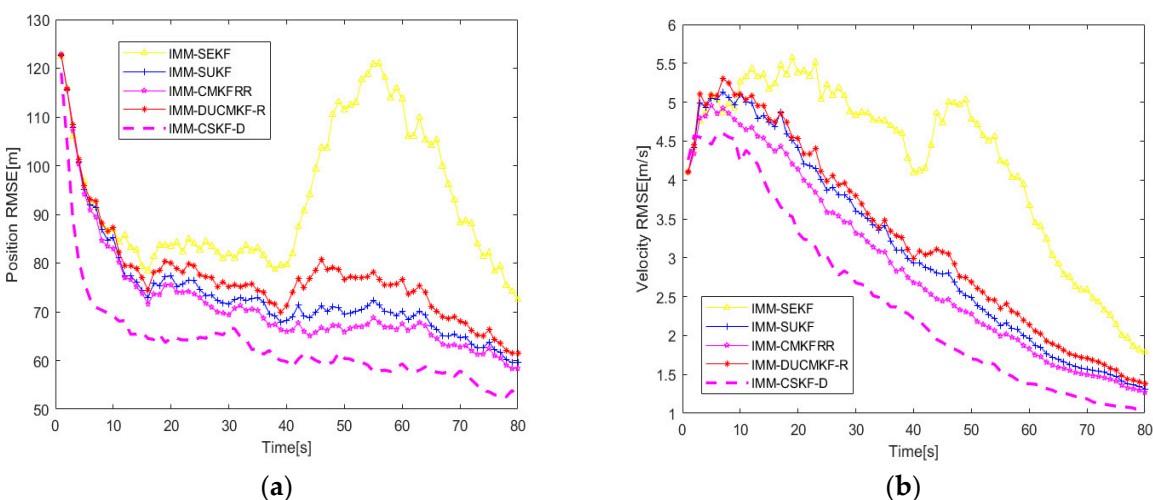

**Figure 8.** RMSEs in IMM (scenario 1). (**a**) RMSE of position. (**b**) RMSE of velocity.

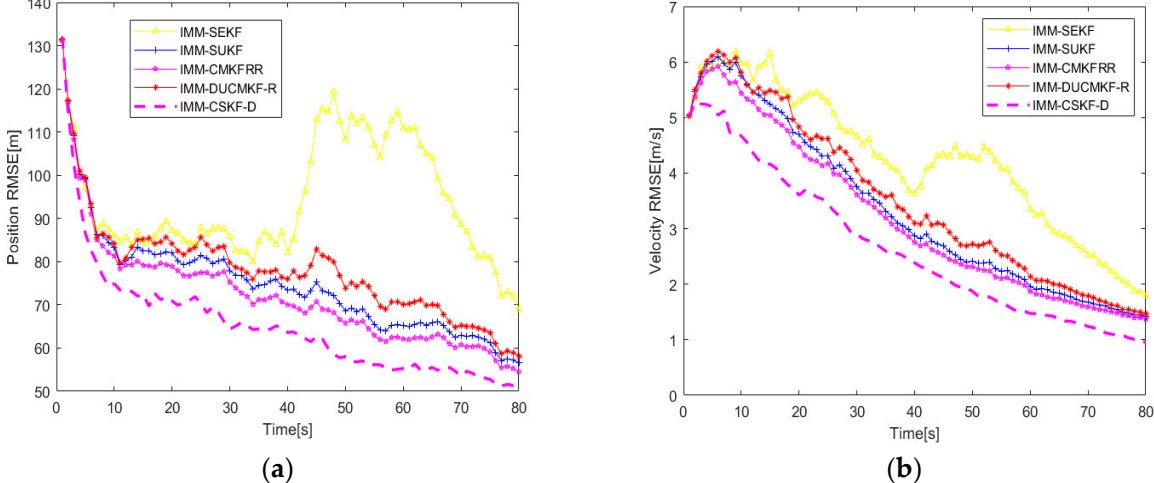

**Figure 9.** RMSEs in IMM (scenario 2). (**a**) RMSE of position. (**b**) RMSE of velocity.

From Figures 8 and 9 and Table 6, it can be seen that our method can maintain good tracking performance when the target makes maneuvering movements. Moreover, IMM-SEKF yields a significant estimation error for the target in the CA motion because it discards high-order terms above the second order during the linearization process. The IMM-CSKF-D algorithm avoids the impact of strong nonlinearity in Doppler measurements through state transition, thereby improving the tracking accuracy and convergence rate.

**Table 6.** Performance comparison in IMM scenarios.

| Measurement Noise Parameters | Method | RMSE of Position (m) | RMSE of Velocity (m/s) |
|---|---|---|---|
| $\sigma_r = 60$ m $\sigma_\theta = 0.6$ deg $\sigma_{\dot{r}} = 0.06$ m/s | IMM-SEKF | 94.67 | 3.97 |
| | IMM-SUKF | 81.70 | 3.08 |
| | IMM-CMKFRR | 78.87 | 2.94 |
| | IMM-DUCMKF-R | 76.34 | 2.79 |
| | IMM-CSKF-D | 66.75 | 2.41 |
| $\sigma_r = 120$ m $\sigma_\theta = 1.2$ deg $\sigma_{\dot{r}} = 0.12$ m/s | IMM-SEKF | 95.74 | 4.37 |
| | IMM-SUKF | 82.94 | 3.74 |
| | IMM-CMKFRR | 79.33 | 3.65 |
| | IMM-DUCMKF-R | 78.05 | 3.48 |
| | IMM-CSKF-D | 69.36 | 2.57 |

In the IMM filtering process, change in the model probability is a crucial factor in determining performance. Therefore, the model probabilities of the five algorithms were simulated to investigate the factor, as shown in Figure 10.

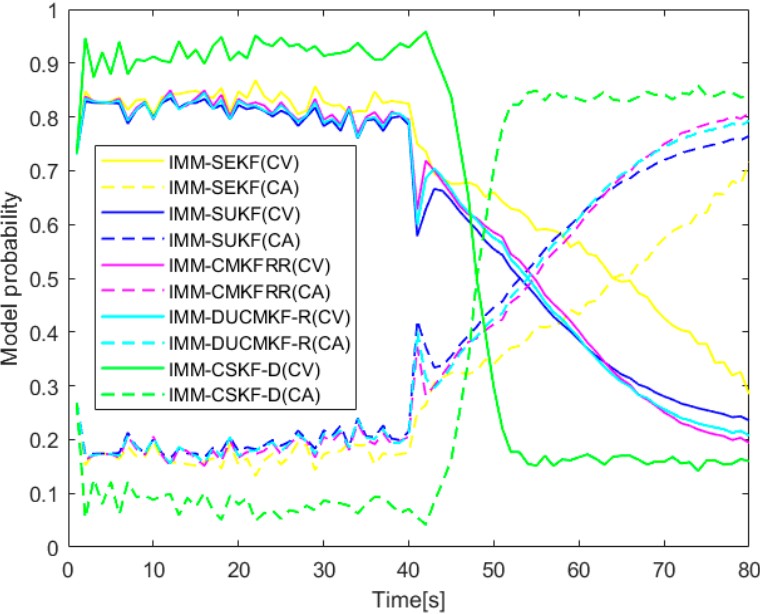

**Figure 10.** Model probability.

As shown in Figure 10, the IMM-CSKF-D algorithm facilitates the model switching speed and enhances the accuracy of the model probability estimation. Figure 10 depicts the probability switching curves of the model; it is evident that the probabilistic evaluation of the IMM-CSKF-D algorithm at each stage of the motion model is better than that of other methods. In terms of selecting a real target model, this algorithm makes the motion of CV and CA more vivid by using more weight, which, in turn, allows for more appropriate model matching for the target's maneuvering changes. As a result, this algorithm can directly estimate the corresponding residual error through linear filtering when calculating

the probabilities of IMM, avoiding errors caused by nonlinearity. In addition, the combination of CSKF-D and IMM can achieve efficient maneuvering target tracking because the CSKF-D algorithm used in the sub-filter possesses higher estimation accuracy.

## 6. Conclusions

This study proposes the IMM-CSKF-D algorithm for Doppler radar target tracking, transforming the nonlinear filtering problem into a linear filtering problem. We developed the CSKF-D algorithm by adding Doppler measurement to the CSKF algorithm measurement equation to solve the problem of strong nonlinearity between the target motion and Doppler measurement. In addition, the UT transformation was used to calculate the statistical characteristics of the process noise converted to the polar coordinate, which improves the calculation accuracy of the noise's statistical characteristics. The model also improves the real-time filtering performance. We combined the CSKF-D algorithm with the IMM to obtain the IMM-CSKF-D algorithm. We aimed to address the target maneuvering motion issues and used this algorithm to solve the Doppler radar maneuvering target tracking problem. Finally, we evaluated the performance and superiority of this model through simulation and comparison with existing methods.

**Author Contributions:** Methodology, X.Z. (Xuanzhi Zhao); Resources, W.Z.; Supervision, Z.L.; Validation, X.Z. (Xian Zhao); writing—review and editing, X.Z. (Xian Zhao). All authors have read and agreed to the published version of the manuscript.

**Funding:** This research received no external funding.

**Data Availability Statement:** The data presented in this study are available from the corresponding author on reasonable request.

**Conflicts of Interest:** The authors declare no conflicts of interest.

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
