# Peer review of "A Method to Track Moving Targets Using a Doppler Radar Based on Converted State Kalman Filtering"

_electronics, doi:10.3390/electronics13081415_

Round 1

Reviewer 1 Report

Comments and Suggestions for Authors

This is an excellent paper making multiple contributions.  The authors converted conventional coordinate systems to Cartesian coordinates, which they argue is better suited to tracking using Doppler radar data.  They introduced a model in which includes Doppler measurements.  The model was tracked using an unscented Kalman filter.  They studied maneuvering targets using multiple model tracking.

Overall the presentation is excellent.

Comments on the Quality of English Language

My only concern is regarding the punctuation of the equations.  It is apparent that when they typeset the manuscript they left a blank line after the equation, which results in an indented "new paragraph" after the equation.  See, for example, line 125 on p. 3.  This problem exists for most of the equations in the paper.  Generally, the equations do not start new paragraphs, and often occur in the middle of sentences.  So this minor change should be fixed.  Also, it is appropriate to punctuate the equations (using periods or commas as needed) in these equations.

Otherwise, the language was very good.  The explanations were clear, well-motivated.  The example were compelling.

Reviewer 2 Report

Comments and Suggestions for Authors

â‘  There are some formatting issues with this file, such as not starting a separate paragraph to explain formula parameters, and starting a separate paragraph for 'where'.
â‘¡ In Section 5.2, there is a significant error in the speed estimation of CSKF-D in the first 10 seconds compared to other methods. Please explain the main reason for this situation in detail.
â‘¢ For Figure 10, there is no more detailed analysis in the paper. It is recommended to draw five different algorithms in one graph, so that the advantages and disadvantages of the different algorithms can be compared more intuitively.
â‘£ It is recommended to clearly describe each experimental scenario and draw a chart to make it more detailed.

Comments on the Quality of English Language

Minor editing of English language required.

Reviewer 3 Report

Comments and Suggestions for Authors

The paper proposes a methodology for target tracking using a doppler radar, adopting Kalman filtering and interacting multiple model techniques. The paper derives the Kalman filtering equations for constant velocity and constant acceleration, describes a method to compute the process noise covariance through unscented transformations, and describes the interacting multiple model equations adopted. The authors compare the proposed method with other state-of-the-art methods through simulations showing that the proposed method is competitive. The paper contains relevant results. I have, however, the following comments:

- The authors should be more clear about what are the different models used in the IMM method. Are they all either CV or CA models, and the only difference is the estimated state?

- The authors should clarify how to select the number of models N, and how are the transition probabilities p_{ij} computed.

- The typesetting of the formulas could be improved, for example equation 13 or line 226.
